# Preliminary Evaluation of Cortical and Medullary Echogenicity in Normal Canine Fetal Kidneys during the Last 10 Days of Pregnancy

**DOI:** 10.3390/vetsci10110639

**Published:** 2023-10-31

**Authors:** Giulia Siena, Francesca di Nardo, Barbara Contiero, Tommaso Banzato, Chiara Milani

**Affiliations:** Department of Animal Medicine, Production and Health, Via dell’Università, 16, 35020 Legnaro, PD, Italy; giulia.siena@phd.unipd.it (G.S.); francescadinardo30@gmail.com (F.d.N.); barbara.contiero@unipd.it (B.C.); tommaso.banzato@unipd.it (T.B.)

**Keywords:** echogenicity analysis, fetal kidneys, pregnancy monitoring, ultrasound, dog

## Abstract

**Simple Summary:**

Variations in the echogenicity of specific areas of the fetal kidneys, cortex and medulla are studied in this work during the last 10 days of pregnancy in bitches. The aim was to measure these variations and to identify any correlation with days before parturition (dbp). For this reason, 10 clinically healthy pregnant bitches (2–8 years old, 8.8–40.3 kg bw) were ultrasonographically evaluated from −10 to 0 days before parturition (dbp). Longitudinal kidney US scanning was performed on the three most caudal fetuses, and quantitative parameters were measured in their renal cortex and medulla, such as the mean gray level (MGL) and SD of a manually drawn region of interest (ROI), using specific (Fiji Image J 1.51h, Java 1.6 0_24 64 bit). software. The parameters affected by dbp only were the cortical SD (C-SD) and cortico-medullary SD (C/M-SD), which decreased as delivery approached, indicating a reduction in the heterogeneity of these structures. As they were only affected by dbp, and not by maternal and litter size, further studies may concentrate on those parameters in order to evaluate their usefulness for pregnancy monitoring and parturition prediction and their diagnostic value for the onset of renal pathologies in neonatal, pediatric and adult dogs.

**Abstract:**

The objective of this study was to assess changes in the echogenicity of the cortex and medulla of canine fetal kidneys in relation to days before parturition (dbp), maternal size and litter size. Monitoring of 10 healthy pregnant bitches (2–8 years old, 8.8–40.3 kg bw) was conducted from −10 to 0 dbp using ultrasound. A single renal sonogram was obtained by scanning in a longitudinal section the three most caudal fetuses. The mean gray level (MGL) and SD of a manually drawn region of interest (ROI) in the renal cortex and medulla were measured using the Fiji Image J software (Image J 1.51h, Java 1.6 0_24 64 bit). A linear mixed model taking into account the maternal size as a fixed effect, dbp and litter size as covariates and the bitch as a random and repeated effect was used. The regression coefficients (b) were estimated. Cortical SD (C-SD) and cortico-medullary SD (C/M-SD) were influenced by dbp, with a significant decrease at the approaching day of parturition (b = 0.23 ± 0.06, *p* < 0.001 and b = 0.5 ± 0.02, *p* = 0.038, respectively). Maternal size had a significant impact on C/M-MGL with differences observed in large-sized (1.95 ± 0.13) compared to small- (1.41 ± 0.10, *p* = 0.027) and medium-sized bitches (1.51 ± 0.09, *p* = 0.016). The C/M-MGL was influenced by litter size, showing a decrease as the number of pups increased (b = −0.08 ± 0.03, *p* = 0.018). C-SD and C/M-SD were exclusively affected by dbp, and not by maternal and litter size. This suggests their potential as valuable parameters, warranting further investigations in future studies.

## 1. Introduction

Parturition timing is of paramount importance for both breeders and veterinarians to minimize neonatal death in the canine species. Planning labor assistance or a cesarean section is essential to reach this goal and ensure bitches’ and pups’ welfare. Pregnancy length is widely variable when calculated from the breeding date, being in a range between 57 and 72 days [1,2,3,4]. For the above reason, formulas based on different fetal parameters assessed using ultrasound (US) were proposed for the prediction of the parturition date in the canine species [5,6,7,8]. The accuracy in the prediction of the parturition date using different methods considering maternal as well as fetal US parameters was studied in different phases of pregnancies and in different breeds and maternal sizes [1,5,7,9,10,11,12,13,14,15,16,17]. Nowadays, the prediction of the parturition day is still challenging, especially when a bitch is presented for the first time for pregnancy monitoring during the last week of gestation and the ovulation day is unknown.

In human and veterinary medicine, US is a useful technique to evaluate pregnancy and fetal organ development. An US image is composed of pixels, with a specific shade of colors from white to black passing through a wide gray scale in relation to the tissue and consequently to the strength of the returning echoes. Due to physical limitations, the human eye can perceive only 10–12 shades of gray [18], and it can only perform a subjective analysis of US images using a qualitative evaluation. In recent decades, the development of new US machines and software for quantitative image analysis has occurred. Quantitative analysis of echogenicity is performed by drawing a region of interest (ROI) selected on the US image. Then, dedicated software is used to measure the different parameters describing the gray scale in the selected ROI [19,20,21,22]. The use of specific software for organ echogenicity analysis is reported to be useful to assess the development of fetal organs in different species [20,21,22,23]. In humans, fetal lung echogenicity is related to the development of the organ, and it has a predictive value for respiratory distress in newborns [23]. In the canine species, the fetal lung-to-liver echogenicity ratio was assessed in the last 3 weeks of pregnancy and it was described as a reliable parameter to assess fetal lung maturity [22]. In humans, the fetal echogenicity of the whole kidney in relation to the fetal liver was studied as a useful diagnostic tool to evaluate normal prenatal kidney development during pregnancy. Fetal kidneys were described as more echogenic during early compared to mid- or late pregnancy and more homogenous in appearance in early compared to mid-pregnancy [20].

In the canine species, the ultrasonographic evaluation of fetal kidneys is possible from day 37–45 post ovulation [5]. Fetal kidney development being assessed using US was first described by Gil et al. (2018) [12], and it was divided into four different phases from the first US visualization of the organ to the parturition day. From 20 to 24 days before parturition (dbp), the fetal kidneys showed a “mushroom” shape, a dilated renal pelvis and a hyperechoic cortex without a cortico-medullary distinction. From 20 to 16 dbp, a cortico-medullary distinction was visible, being hypoechoic in the medulla compared to the cortex, and from 20 to 11 dbp, the cortex became hypoechoic with respect to the previous phases. From day 5 to 1 before parturition, the renal cortex is hypo/isoechoic compared to the liver parenchyma and the renal pelvis is not dilated with a hyperechoic appearance. The cortico-medullary distinction becomes more evident as parturition approaches due to the progressive development of nephrons [12].

The aim of the present study was to evaluate the effect of dbp on the cortical and medullary echogenicity and the ratio between them, objectively measured using dedicated software. Furthermore, it was assessed how maternal size, considering mainly prematernal body weight, and the number of littermates may have an influence on the MGL and SD measured in fetal kidney cortex and medulla.

## 2. Materials and Methods

Ten pregnant bitches (1 Dachshund, 1 Whippet, 1 Norfolk Terrier, 1 Australian Shepherd, 1 French Bulldog, 1 Miniature Bull Terrier, 1 Flat-coated Retriever, 1 Labrador Retriever, 1 Maremmano-Abruzzese Sheepdog and 1 crossbreed dog) were enrolled in the study. They were presented for pregnancy monitoring to the Veterinary Teaching Hospital of the University of Padova in a good state of health, ranging from 8.8 to 40.3 kg in body weight and from 2 to 8 years of age. The number of enrolled bitches was defined using sample size calculation. The dams were divided into 3 different maternal size groups based on the estimation of their pregestational body weight: small (≤10 kg), medium (11–25 kg) and large (≥26 kg).

US examination was performed using an 8–5 MHz convex transducer connected to an US unit (Philips Affiniti 50G, Italy). The US unit was set as follows: the depth was adjusted to optimize the visibility of the fetuses (typically ranging from 3 to 9 cm), and the frequency was set according to the customer settings, with 64 gain and a neutral position of time gain compensation. The bitches were positioned in either dorsal or lateral recumbency following hair clipping, and an US gel was applied on the abdomen. These examinations were performed on two different days, which were not consecutive, and occurred within two distinct time intervals for each patient: the first time interval was from −10 to −5 dbp (referred to as time I) and the second time interval was from −4 dbp to the day of parturition (referred to as time II). On the first monitoring day, recent and remote anamnesis of the dam were collected and a general examination was performed and repeated at each monitoring point. During the last week of pregnancy, a complete blood cell count (CBC) was performed to evaluate the presence of infections or any other hematological alterations or subclinical conditions. US monitoring was scheduled based on the estimated ovulation day. A serum progesterone assay was performed using a fluorescence enzyme immunoassay (FEIA) validated for the canine species [24]. In cases in which the ovulation day was unknown, the estimation of the gestational age was based on fetometric parameters depending on the gestational period in which the US monitoring was performed. Extra-fetal parameters, such as the inner chorionic cavity (ICC), and fetal parameters, like the crown–rump length (CRL) and biparietal diameter (BP), were measured to calculate gestational age or days after delivery using maternal-size-specific formulas [5,6,7,8]. Whenever possible, the most accurate parameters were considered as described elsewhere and measured in order to predict the date of delivery [16,17]. After delivery, the factual dbp were counted backward from the day of parturition (considered as day 0) and the actual US monitoring dates were calculated. Bitches in which the day of ovulation was unknown were excluded when the US monitoring days, counted backward from the parturition day (day 0), were not in the defined time range (−9/0 dbp). The other exclusion criteria for the study were as follows: abnormalities or clinical signs of systemic pathologies in the bitch; bitches who received hormonal drugs (e.g., corticosteroids and sex hormones) within six months prior to the pregnancy under examination; singleton pregnancy; fetuses with fetal heart rate deviations below 160 bpm [10]. Fetuses carrying morphological abnormalities, stillbirth rates higher or equal to 30%, and neonatal deaths of the observed fetuses were also considered as exclusion criteria. A general health check of the examined pups was performed up to 2 weeks after birth. At each US monitoring, the fetal heart rate, placenta and fetal fluids of all the visible fetuses were assessed. 

The fetal kidneys of the three most caudal fetuses in both uterine horns were observed and one sonogram of the most superficial kidney was obtained. The renal sonograms were collected using a longitudinal scan, showing the fetal kidney at its maximum length and with an evident cortico-medullary distinction. The renal sonograms were saved in an 8-bit DICOM format and analyzed using dedicated freeware software, Fiji Image J (Image J 1.51h, Java 1.6 0_24 64 bit, National Institute of Health, 9000 Rockville Pike, Bethesda, Maryland 20892 USA). The mean gray level (MGL) and standard deviation (SD) were measured on the manually drawn areas as regions of interest (ROIs). The ROIs were drawn avoiding the renal poles and artifacts [25,26]. For each image, one area was selected in the renal cortex and another in the medullary region (Figure 1). The MGL is the mean echogenicity of the considered area [27], and it was defined as the ratio between the sum of the intensity of the pixels and the number of pixels present in that area. SD is the standard deviation of the intensity of the pixels contained in the ROI, and it estimates the homogeneity of the pixels in the considered area [27]. The cortical MGL (C-MGL), cortical SD (C-SD), medullary MGL (M-MGL) and medullary SD (M-SD) were assessed. The cortical/medullary ratio (C/M) was also calculated using the obtained MGL and SD values.

A Shapiro–Wilk test was used to assess the normality of the collected data. A statistical analysis of the parameters (C-MGL, M-MGL, C/M-MGL, C-SD, M-SD and C/M-SD) was conducted using a linear mixed model with SAS 9.4 (SAS Institute Inc., Cary, NC, USA). Maternal size was treated as a fixed effect, and it was categorized into three groups depending on the pregestational body weight: small (≤10 kg), medium (11–25 kg) and large (26–40 kg), while days before parturition (−10 to 0 dbp) and litter size (ranging from 3 to12 pups) were included as covariates, and the individual bitch was considered as a random and repeated effect. The output of the statistical analysis includes least squares means (LSM), estimated regression coefficients (b) and standard errors (SE). Subsequent post hoc pairwise comparisons between the different levels of fixed effect were carried out with adjustments made with the use of Bonferroni correction. The raw regressions between the dbp and the considered parameters (C-MGL, M-MGL, C/M-MGL, C-SD, M-SD and C/M-SD) were calculated. The hypotheses of the linear model were assessed graphically by examination of the residuals. Pearson’s correlation coefficient was calculated to explore the existence of any correlation between litter size and maternal body weight measured during the initial clinical examination. The statistical significance was defined as a threshold of *p* < 0.05.

## 3. Results

The US monitoring was conducted twice in 9 out of the 10 included bitches. One bitch was monitored four times (7, 4, 1 and 0 dbp) due to the presence of a fetus displaying fetal anasarca, the collected data of which were excluded from the study. In another case, one bitch gave birth one day later than expected, resulting in an extension of time interval I by one day (−10/−5 dbp). Among the other nine bitches, examinations were performed on the three most caudal fetuses in both uterine horns. The exact ovulation day was unknown in 5 out of 10 female dogs. Among the study participants, four patients were categorized as small-sized, three as medium-sized and three as large-sized bitches. Of these, seven bitches were primiparous while three had previous litters. Four bitches underwent natural parturition, while the remaining six patients required a C-section intervention to complete delivery. The litter size ranged from 3 to 12 pups, and all the examined fetuses were born alive at birth and remained healthy up to 2 weeks of age. 

Hematological analysis and specifically a cell blood count (CBC) was conducted in 8 out of 10 bitches. The results were generally within the normal range, with exceptions related to late pregnancy, including elevated platelet levels, decreased hematocrit and reduced hemoglobin concentration. An increase in the concentration of eosinophils, possibly indicating the presence of intestinal parasites, was found in four out of eight bitches, and no clinical signs were reported by the owners (Table 1).

A total of 126 measurements (*n* = 63 for both the cortical and medullary regions) were acquired from 29 fetuses under examination. There were 30 renal sonograms acquired during the first time point (time I) and 33 during the second time point (time II). All collected data exhibited a normal distribution. The C-MGL was found to be unaffected by dbp and maternal and litter size (*p* = 0.6, *p* = 0.3 and *p* = 0.4, respectively). Conversely, the C-SD was influenced by dbp and exhibited a decrease as delivery approached (b = 0.23 ± 0.06, *p* < 0.001) (Figure 2), while it remained uninfluenced by bitch size and litter size (*p* = 0.1 and *p* = 0.4, respectively).

The M-MGL exhibited an unobservable impact from dbp and litter size (*p* = 0.4 and *p* = 0.3, respectively), whereas for maternal size, the *p*-value was on the threshold (*p* = 0.054). M-SD was not affected by any of the parameters considered: dbp (*p* = 0.5), maternal (*p* = 0.4) or litter size (*p* = 0.9). Cortico-medullary-SD (C/M-SD) was significantly influenced by dbp, displaying a reduction as the day of delivery approached, with a coefficient of b = 0.5 ± 0.02 (*p* = 0.038, Figure 3). Conversely, it remained unaffected by maternal size and litter size (*p* = 0.7 and *p* = 0.2).

The C/M-MGL was influenced by litter size, decreasing as litter size increases (b = −0.08 ± 0.03, *p* = 0.018) (Figure 4) and by maternal size (*p* = 0.01) (Figure 5), with different values between large-sized (1.95 ± 0.13) and medium-sized (1.51 ± 0.09, *p* = 0.016) and between large- and small-sized bitches (1.41 ± 0.10, *p* = 0.027). The C/M-MGL was not affected by dbp (*p* = 0.8). A correlation was found between maternal body weight and litter size (*p* < 0.0001) with a Pearson correlation coefficient of 84.8% (r = 0.848).

## 4. Discussion

To the best of the authors’ knowledge, this is the first study objectively evaluating the cortical and medullary echogenicity parameters (C-MGL, C-SD, M-MGL and M-SD) and their ratios (C/M-MGL and C/M-SD) in canine fetal kidneys and the effect of dbp, maternal and litter size on such parameters. In the present study, C-SD and C/M-SD were affected only by dbp and not by maternal and litter size during the last 10 days of pregnancy. C-SD, describing the heterogeneity of the cortex, decreased as parturition approached. It is the authors’ opinion that this may be due to the development of the glomeruli in the renal cortex, with the renal cortex becoming more homogeneous as the number of these structures increases, as what is reported in human fetuses [27]. C/M-SD also decreased as parturition approached but with a smaller slope compared that for C-SD. This is probably due to the influence on this ratio of M-SD, which was not affected by dbp. The results of this study are in agreement with those reported in human medicine; indeed, in humans, the homogeneity of the renal parenchyma increases with gestational age due to the progressive formation of the nephrons [20,27]. Further histological studies, focused on glomeruli increase and development, considering its relationship with the homogeneity of canine fetal kidneys in the last days of pregnancy are needed to confirm this hypothesis. 

The C/M-MGL, describing the mean echogenicity value of the cortical/medullary ratio, was affected by maternal size, showing higher values in large-sized compared to medium- and small-sized bitches. This parameter was also influenced by litter size, decreasing as the number of pups per litter increased. For a better understanding of these findings, the correlation between maternal and litter size was calculated, finding a correlation of 84.4%. Such results are in accordance with the previously reported literature; indeed, the mean litter size is reported to be highly correlated with maternal body weight, with r = 0.83 in dogs. The number of pups per litter is reported to increase as maternal weight increases [28]. The tissue thickness between the probe and organ, as well as the depth setting used during US examination, can influence the ultrasonographic signal received by the probe, affecting the obtained image and then the results of the image analysis [26,29].

In human medicine, the evaluation of kidney echogenicity is used for monitoring fetal renal maturity and as a diagnostic tool for the detection of congenital nephropathies [20,27,30]. In a recent study, a qualitative evaluation of fetal kidney echogenicity was performed defining renal echogenicity as hyperechoic, isoechoic and hypoechoic to the liver and the spleen. In healthy fetuses, the renal cortex should be isoechoic or hypoechoic compared to the liver or spleen after week 32 of pregnancy. A quantitative evaluation of human fetal kidneys was performed by Hershkovitz et al. (2011) [20] from week 14 to 41 of gestation, studying the echogenicity of the whole fetal kidney parenchyma (cortex, medulla and renal pelvis) using dedicated software, and the results were adjusted according to liver echogenicity. According to what was reported by Devriendt et al. (2013) [27], kidney echogenicity was higher in early compared to mid- and late pregnancy. Moreover, the renal parenchyma was more homogenous in early compared to mid-pregnancy and less homogeneous in mid- compared to late pregnancy. No differences in homogeneity were found between early and late pregnancy. Liver echogenicity showed no differences during the considered pregnancy period (from 14 to 41 weeks of gestation). The calculated kidney/liver ratio index increased as parturition approached and was used with 90% sensitivity and 88% specificity to discriminate between early and late pregnancy [20]. 

In the present study, fetal cortical and medullary echogenicity (C-MGL and M-MGL) were not affected by gestational age during the last 10 days of pregnancy in dogs. These results, in contrast to what is reported in humans, are probably due to the difference in the examined period, which was wider in previous studies (last weeks of pregnancy) compared to the present one, which was focused only on the interval of the last 10 days of pregnancy. Due to the shorter gestational period considered, it was probably not possible to detect changes in the cortical and medullary echogenicity. Further studies considering a longer period of pregnancy should be performed to objectively assess and distinguish possible modifications in cortical and medullary echogenicity and homogeneity in canine fetal development. Furthermore, due to the inclusion of pregnant bitches with a great size difference, it was not possible to standardize the depth parameter for all the collected data, and a ratio including renal echogenicity parameters was needed. Differently from what is reported in humans, in the canine species. fetal liver echogenicity increases significantly during the last week of pregnancy, with respect to the previous 2 weeks [22]. For this reason, in the present study, we standardized the MGL and SD measurements using ratios of cortical and medullary MGL and SD measurements (C/M-MGL and C/M-SD). 

Normal renal and liver echogenicity were determined by comparing US echogenicity values and histological examinations of the same organs in adult dogs and cats [18,31,32]. In adult cats, quantitative evaluation of renal and liver echogenicity was described to be a useful tool to support clinical diagnosis and help early detection and evaluation of the progression of pathological conditions [18]. In adult dogs and cats, in vivo and ex vivo studies were performed to investigate the influence of histopathological lesions on cortical echogenicity [31,32]. The objective evaluation of renal cortex echogenicity showed a low relevance in detecting chronic kidney conditions in adult dogs, whereas it was useful for identifying severe renal damage in cats [32]. In canine fetuses, ultrasonographic kidney development and echobiometrical parameters were seen in healthy fetuses [12,17]. Canine fetal kidney echogenicity was qualitatively assessed and reported to be lower than the maternal kidney echogenicity from week 5 to 8 of pregnancy [33]. 

## 5. Conclusions

In conclusion, C-SD and C/M-SD (homogeneity parameters) were only affected by days before parturition and not by maternal and litter size, decreasing as parturition approaches during the last 10 days of pregnancy. The C/M-MGL was affected by maternal size, showing that cortico-medullary echogenicity seems to vary according to maternal size with higher values in large-sized compared to medium- and small-sized bitches. The evaluation of the echogenicity of the renal cortex and medulla during the last 10 days of pregnancy is time-consuming and affected by the ultrasonographic unit, probe and settings, as well as by the operators’ and patients’ variability. Further studies on different maternal sizes, breeds and body condition scores are needed to investigate the MGL parameters of canine fetal kidneys. However, C-SD and C/M-SD seem to deserve attention in the prediction of the parturition date because they are related just to the days from parturition and not to the maternal and litter size in the canine species. Further studies may be useful to investigate the possible diagnostic value of these parameters collected in the prenatal period for the prediction of parturition, and their relation to the onset of renal pathologies in neonatal, pediatric, juvenile and adult dogs.

## Figures and Tables

**Figure 1 vetsci-10-00639-f001:**
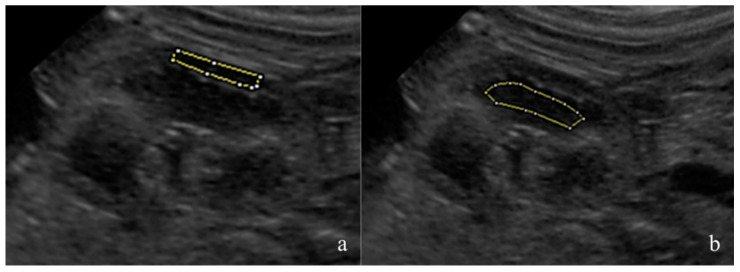
Ultrasonography representing a normal canine fetal kidney in a longitudinal scan showing its maximal length. The image is stored in an 8-bit DICOM format. A manual outline of the region of interest (ROI) within both the renal cortex (**a**) and medullary (**b**) regions were performed by excluding renal poles and any potential artifacts. Subsequently, mean gray level (MGL) and standard deviation (SD) measurements were conducted using Fiji Image J (Image J 1.51h, Java 1.6 0_24 64 bit, National Institute of Health, 9000 Rockville Pike, Bethesda, Maryland 20892 USA).

**Figure 2 vetsci-10-00639-f002:**
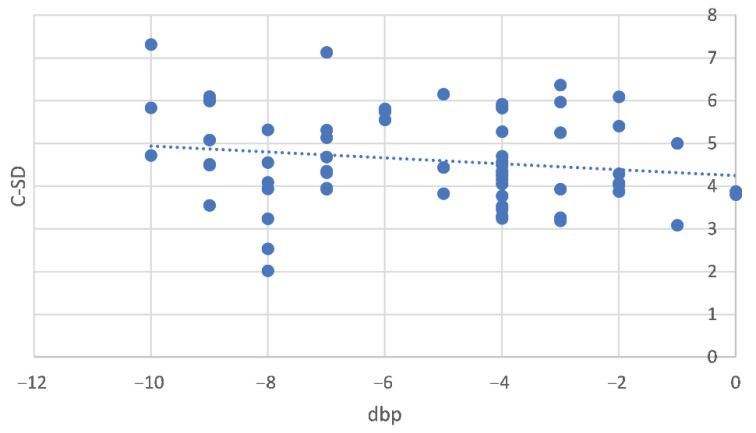
Relationship between standard deviation of cortical echogenicity (C-SD) and the number of days before parturition (dbp) covering the interval from day -10 to day 0 (*n* = 126). Day 0 is the day of parturition. Each blue dot on the graph corresponds to the repeated measure taken from individual fetuses while the dotted line represents the linear regression model. The estimated regression coefficient obtained from the linear mixed model analysis corresponds to b = 0.23 ± 0.06, indicating the degree of change in C-SD with respect to dbp.

**Figure 3 vetsci-10-00639-f003:**
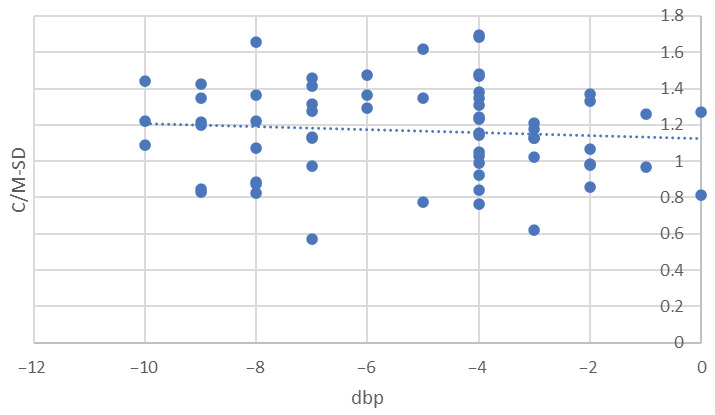
Relationship between standard deviation of cortico-medullary echogenicity ratio (C/M-SD) and the number of days before parturition (dbp) covering the interval from day -10 to day 0 (*n* = 126). Day 0 is the day of parturition. Each blue dot on the graph corresponds to the repeated measure taken from individual fetuses while the dotted line represents the estimated linear regression line determined using a linear mixed model analysis with b = 0.5 ± 0.02. This coefficient quantifies the relationship between C/M-SD and days before parturition (dbp).

**Figure 4 vetsci-10-00639-f004:**
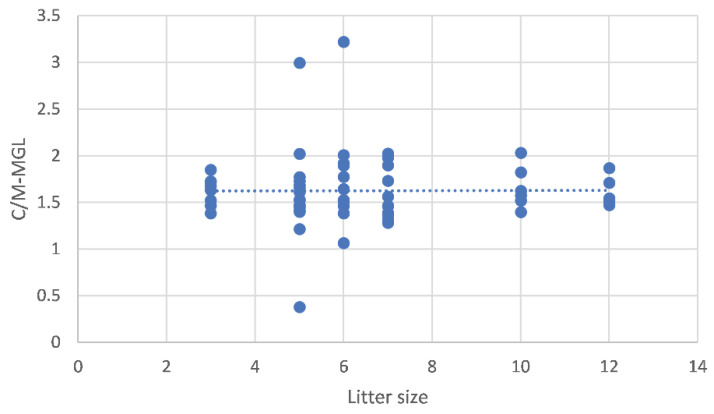
Relationship between mean gray level of cortico-medullary echogenicity ratio (C/M-MGL) and litter size (*n* = 126). Each blue dot on the graph represents the repeated measure taken from individual fetuses while the dotted line represents the linear regression. The estimated regression coefficient obtained using a linear mixed model is b = −0.08 ± 0.03. This coefficient quantifies the relationship between C/M-MGL and litter size.

**Figure 5 vetsci-10-00639-f005:**
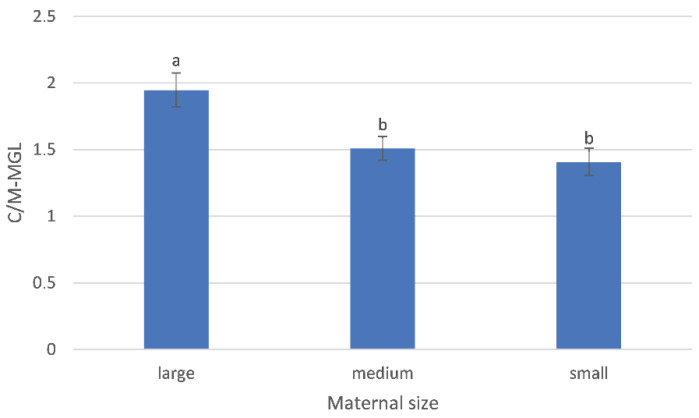
Mean gray level of cortico-medullary echogenicity ratio (C/M-MGL) during the final 10 days of pregnancy in relation to maternal size, categorized as pre-gestational body weight (small ≤ 10 kg, medium 11–25 kg and large ≥ 26 kg) in a total of 126 measurements. C/M-MGL values are represented as bars, least squares mean (mm) ± SE. a, b represent significant differences with *p* < 0.05.

**Table 1 vetsci-10-00639-t001:** Key parameters of patients enrolled in the study, including bitch identification number (n), number of days before parturition (dbp) in which the blood cell count was performed, and various hematological parameters such as hematocrit (HCT, normal range of 38–49.5%), eosinophils (normal range of 130–530/uL), platelet (normal range of 211–384 × 10^3^/uL), hemoglobin (normal range of 13.3–17.2 g/dL) and white blood cells (WBC) (normal range of 8.53–16.61 × 10^3^/µL).

Bitch n	dbp	HCT (%)	Eosinophils (/uL)	Platelet (×10^3^/uL)	Hemoglobin (g/dL)	WBC (×10³/μL)
1	−3	42.8	100	464	14.6	9.08
2	−4	37.1	1080	516	12.3	16.31
3	−4	46.3	50	393	15	14.88
4	−1	37.8	220	173	12.8	7.12
5	−4	34.7	1930	258	11.5	14.05
6	−2	38.7	1100	456	12.8	13.50
7	−3	37.7	860	480	12.7	11.07
8	−4	38.2	157	385	12.7	7.85

## Data Availability

Data are available upon requests.

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
