# Peer review of "Preliminary Evaluation of Cortical and Medullary Echogenicity in Normal Canine Fetal Kidneys during the Last 10 Days of Pregnancy"

_vetsci, 2023, doi:10.3390/vetsci10110639_

Round 1
Reviewer 1 Report
Comments and Suggestions for Authors
In the bitch, determination ''the last 10 days'' of pregnancy is not an easy task. In the materials and methods, the authors did not describe how they determined the last ten days.
The relationship between the results and the pregnancy, particularly the last days of pregnancy, was not described.
It is not clear in the publication what the relationship between echogenicity and pregnancy is. Is this necessary to determine the time of parturition or to determine the state of health of the fetus?
Author Response
Reviewer 1
In the bitch, determination ''the last 10 days'' of pregnancy is not an easy task. In the materials and methods, the authors did not describe how they determined the last ten days.
A: We thank the reviewer for this suggestion and we improved the material and methods section accordingly. As requested, we clarified that US monitoring dates were based on ovulation date or gestational age calculated based on fetal and extra-fetal parameters measured by US (ICC, CRL, BP). The actual delivery date was always recorded and considered as day 0. Factual days before parturition (dbp) were calculated counting backward from actual parturition date and bitches in which the day of ovulation was unknown were excluded when the US monitoring days, counted backward from the parturition day (day 0), were not in the time range defined for our study (-9/0 dbp).
The relationship between the results and the pregnancy, particularly the last days of pregnancy, was not described. It is not clear in the publication what the relationship between echogenicity and pregnancy is. Is this necessary to determine the time of parturition or to determine the state of health of the fetus?
A: We reviewed current literature available on parturition timing parameters available in the canine species (Siena and Milani, 2021). Due to the lack of a parameters useful and accurate in the prediction of parturition date during the last 10 days of pregnancy and previous studies performed in human and veterinary medicine, we investigated in the present preliminary study the variations of renal echogenicity parameters during this specific pregnancy period. As a preliminar evaluation of cortical and medullary echogenicity parameters in normal canine fetal kidneys we focused on the evaluation of the relationship between days before parturition on cortical and medullary echogenicity. Other factors which may influence these echogenicity parameters, as the influence of maternal and litter size, were also investigated. Therefore, in the present study we evaluated these parameters variations in healthy fetuses in order to evaluate the usefulness of studying renal echogenicity for the prediction of parturition date, however this study can also be useful as a baseline for the evaluation of physiological echogenicity values of fetal renal cortex and medulla and future studies about the use of these parameters to determine the canine fetal health status, and prenatal diagnosis of renal pathologies as reported in human medicine.
In order to clarify this point, we modified the conclusion section as follow: “Further studies in different maternal size, breeds and body condition score are needed to investigate MGL parameters of canine fetal kidneys. However, C-SD and C/M- SD seem to deserve attention in the prediction of parturition date because they are related just to the days from parturition and not to the maternal and litter size in the canine species. Further studies may be useful to investigate a possible diagnostic value of these parameters collected in the prenatal period for the prediction of parturition, and their relation to the onset of renal pathologies in the neonatal, pediatric, juvenile and adult dog.”
Reviewer 2 Report
Comments and Suggestions for Authors
This work, entitled “Preliminar evaluation of cortical and medullary echogenicity 2
parameters in normal canine fetal kidneys during the last ten 3 days of pregnancy.”, deal with an interesting topic that should be deepened, with the hope of developing new methods for the prediction of parturition date, especially when progesterone monitoring of the mating is missing (as correctly stated by the author in the paper).
Only minor revisions arose, listed below:
Line 23: change with “the aim of the present study” - in general throghout the manuscript avoid “personal” terms. Authors are invited to revise thisalong the whole manuscript.
Line 40: add “importance” after “paramount”.
Line 61: decide wether to use “grey” or “gray” and use the same term througout the entire manuscript.
Line 169 - 179: it is not clear wether or not you excluded anasarca puppies (and pregnancies) from this study.
Line 197: pay attention to the spaces between “P” and “=” and the related values, they are not the same throughout the manuscript. Check them and use always the same form - choice must be done adhering to authors’ guidelines of the journal.
After line 215 the paragraphs are not justified, pay attention to this.
Line 244: add some commas to make the sentence easier to be read.
Line 283: change “eventual” with “possible”.
Line 298: specify if the sentence is related to adults or fetuses.
Line 305: pay attention with spaces between “-“ and “SD”.
